# Can LLMs Effectively Leverage Graph Structural Information through Prompts in Text-Attributed Graphs, and Why?

**Jin Huang**                                                    *huangjin@umich.edu*
*University of Michigan, Ann Arbor*

**Xingjian Zhang**                                               *jimmyzxj@umich.edu*
*University of Michigan, Ann Arbor*

**Qiaozhu Mei**                                                      *qmei@umich.edu*
*University of Michigan, Ann Arbor*

**Jiaqi Ma**                                                     *jiaqima@illinois.edu*
*University of Illinois Urbana-Champaign*

**Reviewed on OpenReview:** *https://openreview.net/forum?id=L2jRavXRxs*

## Abstract

Large language models (LLMs) are gaining increasing attention for their capability to process graphs with rich text attributes, especially in a zero-shot fashion. Recent studies demonstrate that LLMs obtain decent text classification performance on common text-rich graph benchmarks, and the performance can be improved by appending encoded structural information as natural languages into prompts. We aim to understand why the incorporation of structural information inherent in graph data can improve the prediction performance of LLMs. First, we rule out the concern of data leakage by curating a novel leakage-free dataset and conducting a comparative analysis alongside a previously widely-used dataset. Second, as past work usually encodes the ego-graph by describing the graph structure in natural language, we ask the question: do LLMs understand the prompts in graph structures? Third, we investigate why LLMs can improve their performance after incorporating structural information. Our exploration of these questions reveals that (i) there is no substantial evidence that the performance of LLMs is significantly attributed to data leakage; (ii) instead of understanding prompts as graph structures, LLMs tend to process prompts more as contextual paragraphs and (iii) the most efficient elements of the local neighborhood included in the prompt are phrases that are pertinent to the node label, rather than the graph structure[1].

## 1 Introduction

Large Language Models (LLMs) have gained great popularity for a broad range of applications (Brown et al., 2020; OpenAI, 2023). Recently, there have been a few studies exploring LLMs' effectiveness on graph data, particularly focusing on node classification in text-rich graphs. Consider citation networks as an example, where each node represents a research paper, and each edge indicates a citation relationship between papers. Previous studies have reported decent classification accuracy of LLMs on node-level information alone (Wang et al., 2023; He et al., 2023; Chen et al., 2023). Moreover, recent work also suggests that the incorporation of structural information, e.g., describing the local graph structure and listing the neighboring papers' titles in natural language, can further improve the accuracy (Chen et al., 2023; Guo et al., 2023; Hu et al., 2023).

---

[1]Codes and datasets are at: `https://github.com/TRAIS-Lab/LLM-Structured-Data`

However, the benefit of incorporating structural information can vary across datasets, and the underlying mechanisms remain not fully understood. Indeed, a notable concern arises as most node classification benchmarks have a data cut-off that predates the training data cut-off of LLMs like ChatGPT. This discrepancy raises concerns about data leakage–LLMs may have seen and memorized at least part of the test data of the common benchmark datasets–which could undermine the reliability of studies using earlier benchmark datasets. Furthermore, the variation in performance across different datasets highlights the necessity of a deeper understanding of how LLMs process and benefit from the inclusion of structural information.

To this end, this paper focuses on three concrete questions relevant to the incorporation of structural information into LLMs. First, we seek to understand whether the performance gain of LLMs on common node classification datasets comes from data leakage. Second, we investigate whether LLMs understand the graph structure as topological graphs or contextual paragraphs. Third, we investigate potential reasons why LLMs benefit from structural information.

For the first question, we investigate the extent to which data leakage might artificially inflate the performance of LLMs in Section 3.2. To rigorously measure the data leakage effect, we collect a new dataset, ensuring that the test nodes are sampled from time periods post the data cut-off of ChatGPT (OpenAI, 2022) and LLaMA-2 (Touvron et al., 2023).

For the second question, we investigate how LLMs process structural information - as graphs or paragraphs - by two adversarial settings in Section 3.3. Concretely, we linearize or randomly rewire the ego-graph and then compare the performance between LLMs and Message Passing Neural Networks (MPNNs). We conclude that LLMs understand the input prompt more as linearized paragraphs than as graph-structured data, even though the prompts are intended to represent graph structural information.

For the third question, we probe into two related factors accounting for the improved performance: homophily and the richness of textual node features. In Section 3.4, we examine the impact of homophily (the tendency of similar nodes to connect) on the classification performance of LLMs. Through controlled experiments and correlation analyses, we establish a positive relationship between the local homophily ratio and the prediction accuracy of LLMs. It also implies that homophily is important because it enables the local neighborhood to offer relevant phrases, thereby enhancing classification performance. In Section 3.5, we investigate the circumstances under which structural information provides minimal improvement, by varying the richness of textual information of the target node. Therefore, we discover that LLMs benefit from structural information primarily when the target node lacks sufficient textual information.

Our key findings are summarized as follows. (i) There is no strong evidence that data leakage is a major factor contributing to the performance of LLMs on node classification benchmark datasets. (ii) LLMs understand the prompts as linearized paragraphs,rather than as explicit graph structures. (iii) The most efficient elements of the local neighborhood included in the prompt are phrases that are pertinent to the node label, rather than the graph structure.

Overall, this study investigate the underlying mechanism of how LLMs process graph data. Our findings clarify that data leakage does not significantly boost LLMs' performance on node classification tasks. While the performance does not come from data leakage, our study finds that LLMs do not benefit from the topological structure of the ego-graph either. On the contrary, we discover that the effectiveness of incorporating structural information into the prompts largely depends on the phrases related to the node label. This insight opens new avenues for utilizing LLMs in graph-based applications and for further exploration into advanced methods of incorporating structured data into LLM frameworks.

## 2 Related Literature

**Data leakage in LLMs.** Data leakage in LLMs has become a focal point of discussion due to the models' intrinsic ability to memorize training data. As demonstrated by Carlini et al. (2022), LLMs can emit memorized portions of their training data when appropriately prompted, a phenomenon that intensifies with increased model capacity and training data duplication. While memorization is inherent to their function, it raises serious security and privacy concerns. A study by Carlini et al. (2021) shows that extraction attacks can recover sensitive information such as personally identifiable information (PII) from GPT-2 (Radford et al.,

2019). This capability to store and potentially leak personal data is further explored by Huang et al. (2022), confirming that although the risk is relatively low, there is a tangible potential for information leakage. Specifically, Carlini et al. (2022) show that the 6 billion parameter GPT-J model (Wang & Komatsuzaki, 2021) memorizes at least 1% of its training dataset. Furthermore, the issue of data leakage complicates the evaluation of these models. As highlighted by Aiyappa et al. (2023), the closed nature and continuous updates of models like ChatGPT make it challenging to prevent data contamination, affecting the reliability of evaluation on LLMs in various applications. In node classification tasks, a concurrent work by Chen et al. (2023) observe that a specific prompt alteration significantly improved performance on OGBN-ARXIV, raising concerns about potential test data leakage. In this work, we take a rigorous approach by curating a new dataset for node classification tasks, which is explicitly designed to address the data leakage issues in existing benchmarks.

**LLMs on Text-Rich Graphs.** Recently there has been a series of research on LLMs and text-rich graphs. He et al. (2023) propose a method where LLMs perform zero-shot predictions along with generating explanations for their decisions, which are then used to enhance node features for training Message Passing Neural Networks (MPNNs) (Gilmer et al., 2017) to predict node categories. Chen et al. (2023) extend the work of He et al. (2023) by using LLMs both as feature enhancers and as predictors for node classification. They offer several observations such as Chain-of-thoughts is not contributing to performance gains. Guo et al. (2023) perform an empirical study on using LLMs to solve structure and semantic understanding tasks. More recently, Ye et al. (2023) propose InstructGLM for the instruction tuning of LLMs, like LLaMA (Touvron et al., 2023), for node classification tasks. Zhao et al. (2023) introduces GraphText to encode graph into natural languages using a graph-syntax tree. Our study differs with this line of research in terms of the motivation: while we are using text-rich graph datasets as a testbed, our primary goal is to gain deeper understanding of LLMs' capability of processing the graph modality, instead of leveraging LLMs to better solve node classification tasks.

A few previous studies observe that LLMs have decent classification performance on text-rich graphs based on node-level feature only (He et al., 2023; Chen et al., 2023; Guo et al., 2023; Hu et al., 2023). For example, predicting the category of a paper based on its title and abstract. These investigations typically involve using manually crafted templates to describe neighborhood information around the target node hop-by-hop. Furthermore, several studies have noted that integrating structural information can moderately enhance prediction accuracy. Hu et al. (2023) observe that "Incorporating structural information can slightly improve the performance of GPT in node-level tasks". Similarly, Chen et al. (2023) find that including neighborhood summarization could lead to performance gains, verified across four datasets. Guo et al. (2023) report a significant 10% improvement in prediction accuracy on OGBN-ARXIV when employing 2-hop summarization, compared to using only the target node. Moreover, they find that 2-hop summarization outperforms 1-hop summarization. However, these studies primarily focus on empirical observations without delving into the underlying reasons behind LLMs' improved performance with added structural information.

**Homophily in graph learning.** The concept of homophily (McPherson et al., 2001), which describes the tendency of nodes to form connections with similar nodes, plays an important role in the effectiveness of various graph learning methods (Zhu et al., 2020; Halcrow et al., 2020; Maurya et al., 2021; Lim et al., 2021). The principle of homophily enables MPNNs to smooth node representations by aggregating features from their likely similarly-labeled neighboring nodes. This aggregation process is particularly effective in various types of real-world graphs, such as political networks (Knoke, 1990), and citation networks (Ciotti et al., 2016). Despite its benefits, the reliance on homophily presents a challenge: MPNNs tend to underperform in graphs characterized by heterophily, where connected nodes are likely to differ in properties or labels (Zhu et al., 2020). Notably, the impact of homophily on the integration of structured data into LLMs remains an open area for exploration.

Table 1: Prompt styles and their corresponding templates. For the style "$k$-hop title+label", we only include the labels for neighbor nodes in training set or validation set. The "attention extraction" and "attention prediction" are respectively the two steps of prompts for the $k$-hop attention strategy.

| Prompt Style | Prompt Template |
|---|---|
| Zero-shot | Abstract: \<abstract>\nTitle: \<title>\nDo not give any reasoning or logic for your answer. \nAnswer: \n\n |
| Zero-shot CoT | Abstract: \<abstract>\nTitle: \<title>\nAnswer: \n\nLet's think step by step. \n |
| Few-shot | Abstract: \<few-shot abstract>\n... \nAnswer: \n\n\<few-shot label>\n... (more few-shot examples)\nAbstract: \<abstract>... \nAnswer: \n\n |
| $k$-hop title, $k$-hop title+label | Abstract: \<abstract>\nTitle: \<title>\nIt has following neighbor papers at hop 1:\n
Paper 1 title: \<paper 1 title>\nLabel: \<paper 1 label>\nIt is linked to paper \<list of papers linked to paper 1>\n
Paper 2 title: \<paper 2 title>\nLabel: \<paper 2 label>\nIt is linked to paper \<list of papers linked to paper 2>\n
... (more 1-hop neighbors)\n
It has following neighbor papers at hop 2:\n
... (more 2-hop neighbors)\n
Do not give any reasoning or logic for your answer. \nAnswer: \n\n |
| $k$-hop attention. Step 1: Attention extraction | The paper of interest is \<title>. Please return a Python list of at most \<k> indices of the most related papers among the following neighbors, ordered from most related to least related. If there are fewer than \<k> neighbors, just rank the neighbors by relevance. The list should look like this: [1, 2, 3, ...]\n1: \<neighbor title 1>\n... (more 1-hop neighbors) \n |
| $k$-hop attention. Step 2: Attention prediction | Abstract: \<abstract>\nTitle: \<title>\nIt has following important neighbors, from most related to least related:\n(more neighbors chosen by attention)\nDo not give any reasoning or logic for your answer. \nAnswer: \n\n |
| Linearized $k$-hop title, $k$-hop title+label: | Abstract: \<abstract>\nTitle: \<title>\n
\<paper 1 title>\nLabel: \<paper 1 label>\n
\<paper 2 title>\nLabel: \<paper 2 label>\n
... (more 1-hop neighbors)\n
... (more 2-hop neighbors)\n
Do not give any reasoning or logic for your answer. \nAnswer: \n\n |

## 3 Why Can LLMs Benefit from Structural Information?

### 3.1 Research Questions

In this section, we aim to gain a deeper understanding of three central questions. First, does the classification performance of LLMs on common node classification datasets come from data leakage? Second, do LLMs understand the prompts as graph structures? Third, what factors contribute to LLMs benefiting from structural information? To ground our study, we experiment with the ChatGPT API and LLaMA-2-7B model on node classification datasets that have textual node features.

For the first question, we investigate the potential impact of data leakage in Section 3.2. For the second question, we first investigate whether LLMs understand the input prompts as graph structure in Section 3.3. Building on this insight that LLMs actually understand the input prompts more as a linearized paragraph, we explore two contributing factors of LLMs performance after incorporating structural information. In Section 3.4, we explore the influence of homophily on LLMs performance. In Section 3.5, we assess how the feature richness of target node affects the performance of LLMs.

### 3.2 Data Leakage as a Potential Contributor of Performance

While LLMs have achieved decent classification performance on common node classification datasets, there is a risk that the performance of LLMs is artificially inflated by data leakage (He et al., 2023; Guo et al., 2023; Hu et al., 2023). Note that most node classification benchmark datasets have a data cut-off at 2019 (see Table 9 in Appendix B.1), but ChatGPT (the model used throughout the paper is `gpt-3.5-turbo-0613`) was trained

Table 2: Statistics of OGBN-ARXIV and ARXIV-2023 datasets. Both represent directed citation networks where each node corresponds to a paper published on arXiv and each edge indicates one paper citing another. The metrics In-Degree/Out-Degree, Average Degree, and Published Year are presented for test nodes.

| Dataset | Full Dataset | | Test Set | | |
|---|---|---|---|---|---|
| | #Nodes | #Edges | In-Degree/Out-Degree | Average Degree | Published Year |
| OGBN-ARXIV | 169343 | 1166243 | 1.33/11.1 | 12.43 | 2019 |
| ARXIV-2023 | 33868 | 305672 | 0.16/10.6 | 10.76 | 2023 |

on data up to September 2021 (OpenAI, 2023) and LLaMA-2 was trained on data up to September 2022 [2]. While the training datasets of ChatGPT and LLaMA-2 are not publicly available, given the widespread of these datasets on the internet and the enormous training corpus of ChatGPT and LLaMA-2, it is reasonable to worry about the data leakage issue on these datasets.

To this end, we curate a new node classification dataset, ARXIV-2023, which is designed to resemble another widely-used dataset, OGBN-ARXIV (Hu et al., 2020) as much as possible except that the test nodes are chosen as arXiv Computer Science (CS) papers published in 2023. With the new dataset, we can rigorously investigate the influence of data leakage by comparing the LLM performance between ARXIV-2023 and OGBN-ARXIV.

**Dataset collection.** While, ideally, we should curate the new dataset by simply extending OGBN-ARXIV by including new papers, this is practically challenging for a couple of reasons. In particular, OGBN-ARXIV represents arXiv CS papers in the Microsoft Academic Graph (MAG) until 2019 (Hu et al., 2020), where MAG is a heterogeneous graph representing scholarly communications (Wang et al., 2020). Unfortunately, MAG and its APIs were retired in 2021 and no subsequent data is available.[3] Furthermore, the pipeline to collect and construct MAG is not publicly released. Consequently, we develop our own data collection pipeline to create ARXIV-2023. Specifically, we first sample test nodes from arXiv CS papers published in 2023, and then gather papers within a 2-hop of these test nodes to create a citation network. More details about collection can be found in Appendix B.2.

**Comparison between arxiv-2023 and ogbn-arxiv.** As can be seen in Table 2, ARXIV-2023 and OGBN-ARXIV share great similarities in their network characteristics, with consistent in-degree/out-degree pointing to analogous citation behaviors. ARXIV-2023 shows a lower average in-degree in the test set, which is likely because the test papers in ARXIV-2023 are new and have not had much time to accumulate citations. Additionally, Figure 1 illustrates that the label distributions of the two datasets are comparable. A notable trend from ARXIV-2023, in alignment with arXiv statistics,[4] indicates a rise in AI-related categories like ML, LG, CL, reflecting the current academic focus.

Furthermore, we compare the performance of MPNNs on the two datasets. As can be seen from the two bottom rows in Table 3, we observe that the performance metrics for MPNNs (GCN and SAGE) across both datasets are closely matched, suggesting that both datasets present comparable challenges for classification. For a more comprehensive setting of MPNNs, one can refer to Appendix C.

**LLM performance on arxiv-2023 and ogbn-arxiv.** To test LLMs on ARXIV-2023 and OGBN-ARXIV, we create prompts that encode both the textual features and the local graph structure of a target node in natural language as in Table 1, and then request ChatGPT API or LLaMA-2-7B to make predictions for the target node[5]. The prompt for each node is formulated in one of several styles, as we introduce in details in Appendix A. Additionally, a fixed dataset-level instruction is attached to the prompt when the prompt is sent to all LLMs. The dataset-level instructions are listed in Table 8, Appendix A.

---

[2]https://github.com/facebookresearch/llama/blob/main/MODEL_CARD.md
[3]https://www.microsoft.com/en-us/research/project/microsoft-academic-graph/
[4]https://info.arxiv.org/help/stats/2021_by_area/index.html
[5]We have used `gpt-3.5-turbo-0613` and `LLaMA-2-7B-chat` for throughout the experiments, unless stated otherwise.

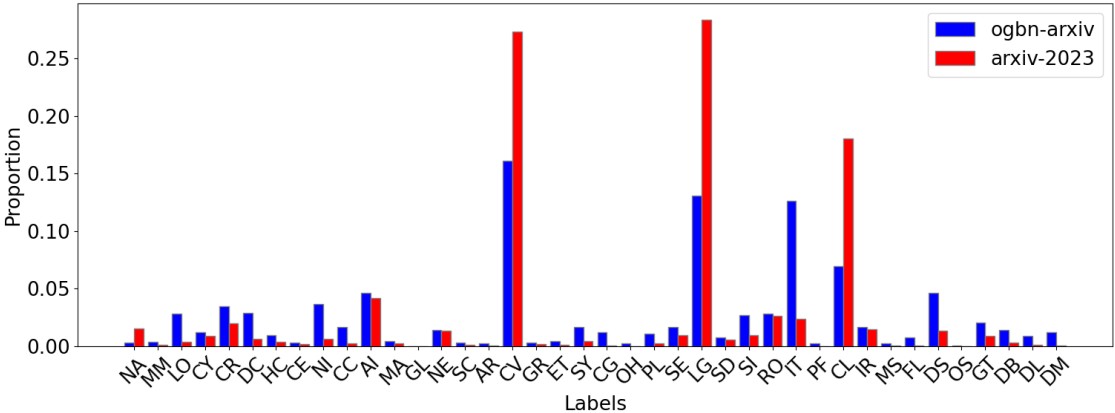

Figure 1: Proportional distribution of labels in OGBN-ARXIV and ARXIV-2023 datasets. Each label represents an arXiv Computer Science Category.

Table 3: Comparison between ChatGPT's performance on OGBN-ARXIV and ARXIV-2023. Best results in prompting methods are **in bold**. 1-hop attention means attention extraction and prediction over 1-hop neighbors

| Rich context | | | Scarce context | | |
|---|---|---|---|---|---|
| Prompt style | OGBN-ARXIV | ARXIV-2023 | Prompt style | OGBN-ARXIV | ARXIV-2023 |
| Zero-shot | 74.0 | 73.5 | Zero-shot | 69.8 | 66.6 |
| Few-shot | 72.9 | 73.6 | 1-hop title | 72.3 | **70.7** |
| Zero-shot CoT | 71.8 | 73.7 | 1-hop title+label | **74.3** | 70.4 |
| 1-hop title+label | **75.1** | **73.8** | 2-hop title | 71.3 | 68.9 |
| 2-hop title+label | 74.5 | 73.2 | 2-hop title+label | 74.2 | 68.5 |
| 1-hop attention | 74.7 | 73.7 | 1-hop attention | 71.3 | 69.6 |
| GCN | 75.4 | 70.3 | GCN | 74.8 | 70.3 |
| SAGE | 75.0 | 70.9 | SAGE | 74.4 | 69.1 |

If data leakage is a major contributor of performance on OGBN-ARXIV, we would expect the performance drop of LLMs between OGBN-ARXIV (may have leakage problem) and ARXIV-2023 (leakage-free) should be **significantly greater than** the drop on MPNNs on two datasets. This is because LLMs may benefit from their memory on OGBN-ARXIV, but this advantage is not likely on ARXIV-2023. However, as shown in Table 3, we observe **the contrary**: the performance drop of ChatGPT between OGBN-ARXIV and ARXIV-2023 is less than the drop on MPNNs on two datasets (1.3% compared to 5.1% in rich context, 3.6% compared to 4.5% in scarce context). For LLaMA-2-7B, we also observe no significant drop compared to MPNNs (6.7% compared to 5.1% in rich context, 4% compared to 4.5% in scarce context. See Appendix D.2, Table 10). This means that LLMs actually generalize well to leakage-free data.

To conclude, the observed results neither offer clear evidence in favor of data leakage nor does it advocate that data leakage predominantly improves LLM's performance. Instead, LLM's consistent performance across both datasets stresses its resilience and ability to generalize across varying distribution domains.

### 3.3 Are LLMs Treating the Prompts as Graphs or Paragraphs on Text-attributed Graphs

?

Previous works design templates to encode the ego-graph around the target node into natural languages by listing neighboring nodes' text information (Chen et al., 2023; Hu et al., 2023) or giving edge indexes (Tang et al., 2023), adjacency lists (Das et al., 2023) or neighborhood summary (Guo et al., 2023). It is implied that some previous works hold that LLMs may benefit from describing the explicit graph structures. However, it

is unclear whether LLMs actually understand the prompt as intended. Thus, we ask the question: do LLMs treat the input prompt as a graph, or merely as a paragraph of text with augmented keywords? This question is fundamental in understanding the performance gain after incorporating structural information. If LLMs just process the prompt as a paragraph consisting of neighboring papers' titles, then the performance gain can only come from the extra phrases included in neighboring nodes, instead of the topological information. We investigate this question by two adversarial experiments:

1. *Linearize ego-graph.* We create a linearized version of the graph-structured prompts by only keeping all neighbors' text attributes in the prompts. We then test the linearized prompts against the graph-structured prompts. Templates of both linearized and graph-structured prompts are shown in Table 1.

2. *Rewire ego-graph.* We randomly rewire the ego-graph by different strategies. Then we compare the performance of MPNNs and LLMs under each strategy.

Table 4: Comparison of classification accuracy between graph-structured prompts and linearized prompts on ARXIV-2023. The performance difference is negligible between two prompt styles, implying that LLMs are treating prompts simply as linearized paragraphs.

| Prompt Style | 1-hop title | 1-hop title+label | 2-hop title | 2-hop title+label | 1-hop attention |
|---|---|---|---|---|---|
| Graph-structured prompt | 71.1 | 67.3 | 69.6 | 67.7 | 74.7 |
| Linearized prompt | 70.9 | 69.1 | 70.8 | 68.2 | 74.9 |

**Adversarial Experiment on Linearizing the Ego-graph.**   To verify whether prompts are processed as linearized paragraphs, we create a linearized version of $k$-hop title, $k$-hop title+label and $k$-hop attention by stripping away descriptive text about the ego-graph structure from the original prompt. For instance, in the original $k$-hop title, a typical introduction like "It has the following neighbor papers at hop $k$:" is removed. In the linearized version, we condense this to a single paragraph comprising all neighbors' titles, as shown in the last row of Table 1. We then compare the performance of the graph-structured and linearized prompts on ChatGPT. As indicated in Table 4, the results reveal minimal performance disparity across all five prompt styles after linearization. A notable finding is that in the 1-hop attention category, even the elimination of text specifying neighbors' importance ranking ("It has the following important neighbors, from most related to least related") has little impact. These findings imply that LLMs tend to process prompts in a linearized paragraph format.

**Adversarial Experiment on Rewiring the Ego-graph.**   We evaluate both MPNNs and LLMs in an adversarial setting where the ego-graph is randomly rewired using three different strategies: "Random" keeps 1-hop neighbors and randomly connects 2-hop neighbors to 1-hop neighbors; "Extreme" retains 1-hop neighbors and connects all 2-hop neighbors to a random 1-hop neighbor; "Path" randomly connects all 1-hop neighbors into a path, as shown in Figure 2. For LLMs, we use the *2-hop title* as the prompting style. The results in Table 5 show that LLMs experience a notably smaller average performance drop compared to MPNNs when subjected to ego-graph rewiring. The average drop of LLMs is -0.25%, compared to -5.55% on MPNNs. When the prompt encompasses all neighborhood textual information, surprisingly the explicit ego-graph structure has minimal impact on LLMs' performance. LLMs' resilience to changes in ego-graph structures suggests that LLMs may process prompts more like paragraphs than graphs.

In summary, our study suggests that LLMs interpret inputs more as contextual paragraphs than as graphs with topological structures. By two adversarial experiments, we show that neither linearizing nor rewiring ego-graph has significant impact on the classification performance of LLMs.

### 3.4   Impact of Homophily on LLMs Classification Accuracy

In the preceding section, we established that LLMs understand the input prompts more as a linearized paragraph. Thus, we hypothesize that relevance of these paragraph prompts to the target node might influence the predictive accuracy of LLMs. Here, we investigate a related factor contributing to the improved

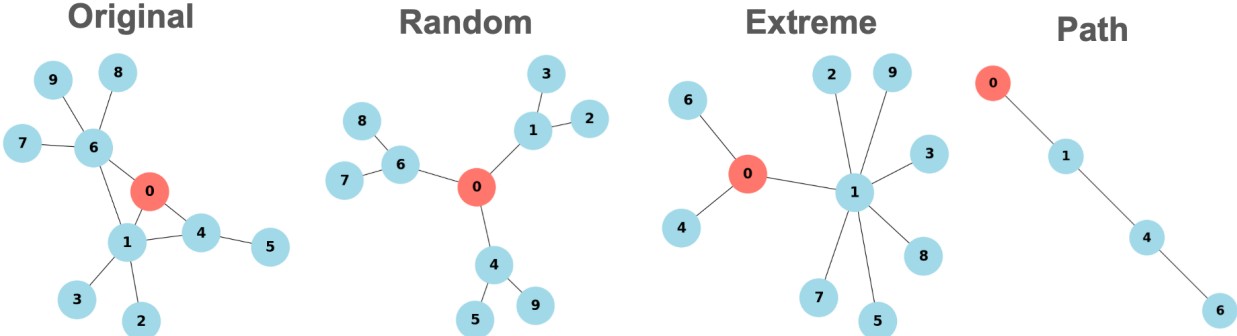

Figure 2: Example of rewiring ego-graphs for node 0. Three rewiring strategies are evaluated: "random" keeps 1-hop neighbors and randomly connect 2-hop neighbors to 1-hop neighbors; "extreme" keeps 1-hop neighbors and connects all 2-hop neighbors connect to a random 1-hop neighbor; "Path" randomly connects 1-hop neighbors as a path.

Table 5: Performance of MPNNs and LLMs under adversarial settings, where we randomly rewire the ego-graph. The last column "Average Drop" denotes the average change of performance in three rewiring strategies compared to the original graph. A larger drop in the last column indicates a greater influence from rewiring the ego-graph.

| Dataset | Model | Original | Random | Extreme | Path | Average Drop |
|---------|-------|----------|--------|---------|------|--------------|
| Cora | GCN | 84.6 | 85.2 | 82.7 | 78.4 | -2.5 |
| | SAGE | 84.1 | 83.8 | 82.7 | 78.5 | -2.4 |
| | ChatGPT | 68.3 | 68.6 | 69.4 | 68.6 | 0.6 |
| | LLaMA-2-7B | 52.2 | 53.5 | 53.5 | 54.1 | 1.5 |
| Arxiv | GCN | 74.7 | 65.2 | 68.2 | 59.3 | -10.5 |
| | SAGE | 75.2 | 71.0 | 67.0 | 66.2 | -7.1 |
| | ChatGPT | 71.5 | 71.4 | 71.1 | 69.0 | -1 |
| | LLaMA-2-7B | 48.3 | 48.9 | 49.2 | 44.4 | -0.8 |
| Arxiv-2023 | GCN | 69.8 | 64.5 | 66.1 | 59.9 | -6.3 |
| | SAGE | 68.1 | 66.5 | 63.4 | 60.8 | -4.5 |
| | ChatGPT | 68.7 | 69.0 | 69.1 | 67.8 | 0 |
| | LLaMA-2-7B | 47.6 | 48.1 | 47.9 | 41.3 | -1.8 |

performance after incorporating structural information, particularly homophily. Homophily, the tendency of nodes with similar characteristics to connect, is foundational for many MPNNs. In fact, the degree of homophily in a dataset often correlates with the efficacy of MPNNs in classification tasks (Zhu et al., 2020; 2021; Lim et al., 2021; Maurya et al., 2021). Given this significance, it becomes imperative to explore if and how homophily impacts the efficacy of LLMs in similar classification contexts, drawing potential parallels or contrasts with MPNN behaviors.

Since LLM performs node-wise prediction over the neighborhood surrounding the target node, we use *local homophily ratio* (Loveland et al., 2023) to measure the degree of homophily with respect to the target node. For a prompt to predict the category of a target node, the local homophily ratio is defined as the fraction of neighbors sharing the same groundtruth label as the target node over the total number of neighbors included in the prompt. Intuitively, a higher local homophily ratio signals scenarios where a node is surrounded by a greater proportion of neighbors from the same category.

**The neighbor dropping experiment.** We design a controlled experiment to demonstrate the effect of local homophily ratio on prediction accuracy. We gradually drop neighbors in three different ways: a) drop the neighbors with same label as the target node; b) drop the neighbors with different label as the target node; and c) drop neighbors randomly. We include details about the neighbor dropping strategies in Appendix E. The experimental results are shown in Figure 3, where we observe an evident trend: as we selectively remove neighbors sharing the same labels, there's a decrease in prediction accuracy. Conversely, discarding

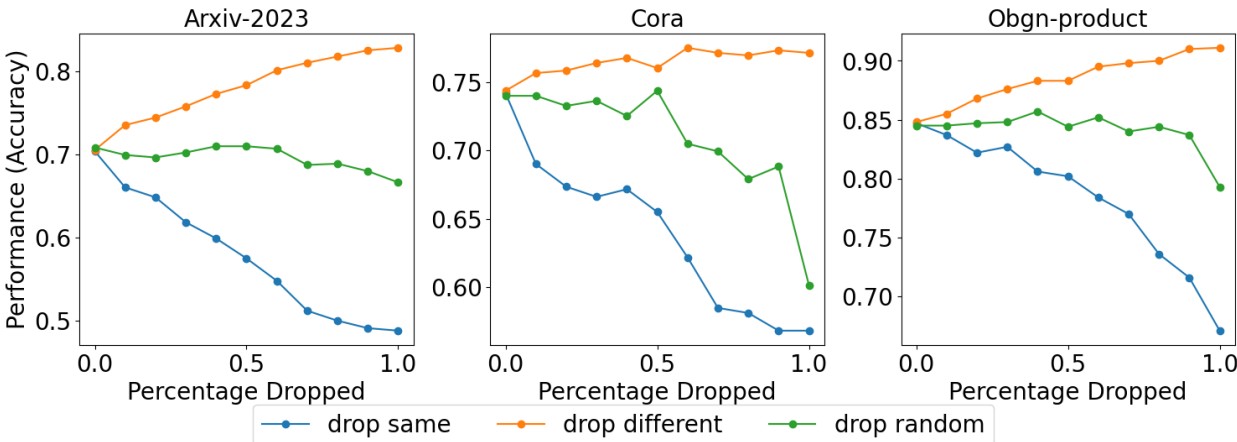

Figure 3: Performance comparison of dropping neighbors using different strategies across ARXIV-2023, CORA, and OGBN-PRODUCT datasets. Three dropping strategies are evaluated: "drop same" removes neighbors with the same label as the target node; "drop different" removes neighbors with different labels as the target node; and "drop random" randomly selects neighbors for removal. When percentage is 1, "drop same" strategy drops all same-label neighbors but preserves all different-label neighbors, and "drop different" strategy drops all different-label neighbors but preserves all same-label neighbors. Details about the strategies are stated in Appendix E.

Table 6: Point biserial correlation between local homophily ratio and prediction correctness across five datasets (p-values in brackets). Point biserial correlation ranges between $[-1, 1]$, where a value of 1 indicates a perfect positive relationship. A higher correlation value indicates that the local homophily ratio and prediction correctness are more positively related.

| Prompt Style | ogbn-arxiv | cora | pubmed | arxiv-2023 | ogbn-product |
|---|---|---|---|---|---|
| Zero-shot | 0.440 (0.000) | 0.070 (0.106) | 0.278 (0.000) | 0.367 (0.000) | 0.387 (0.000) |
| 1-hop title+label | 0.518 (0.000) | 0.222 (0.000) | 0.443 (0.000) | 0.481 (0.000) | 0.560 (0.000) |

neighbors with different labels leads to an increase in accuracy. This selective dropping inherently modifies the local homophily ratio within the prompts. The results show that accuracy of predictions made by LLMs is positively related to local homophily ratio. A similar experiment adding neighbors instead of dropping neighbors in Appendix E also shows a similar trend. Combine this observation with the premise that LLMs actually understand the prompts as a paragraph, we propose that the significance of homophily is intrinsic. This is because LLMs benefit from structural information only when the neighborhood is homophilous, meaning the neighbors contain phrases related to the groundtruth label of the target node. Conversely, diminishing the local homophily ratio by excluding such neighbors can adversely affect prediction accuracy.

**Correlation study.** Building on the insights from the dropping neighbors experiment, we further investigate the relationship between local homophily ratio and the prediction correctness across different datasets. Each node possesses two key attributes: a) its local homophily ratio, which is a continuous random variable in $[0, 1]$, and b) its prediction correctness, which is a binary random variable (0 indicating an incorrect prediction and 1 indicating a correct prediction). To quantify the correlation between these two attributes, we employ the point biserial correlation method (Kornbrot, 2014). This correlation coefficient ranges between -1 and 1, where a value of 1 signifies a perfect positive relationship. The results of our analysis across five datasets are detailed in Table 6.

For the CORA dataset, we observe no significant correlation when only the title is used in prompts. However, a positive correlation emerges when neighbors are included alongside the title. This suggests that the more homophily is incorporated into the prompt, the more accurate the prediction becomes.

For the other datasets, a positive correlation is evident in both the zero-shot and 1-hop title+label settings. In Table 6, zero-shot prediction (the one that doesn't use structural information at all) also showed high correlation with the homophily ratio of the node. This suggests a complicated mechanism for LLMs to perform better on homophilous nodes: those nodes are easier to be classified in the first place; the added structural information has some additional contributions. Homophilous nodes are easier to classify potentially because nodes that are not homophilous often blend various topics, which makes predicting their category more challenging than homophilous nodes.

In summary, our findings underline the critical role of homophily in influencing LLM's text classification performance with extra structural information. The experiments and analyses consistently point to a positive relationship between local homophily ratio and prediction correctness. It further suggests that homophily matters because, only with homophily, the local neighborhood can provide relevant phrases, thus improving the overall performance.

### 3.5 Influence of Structural Information on LLMs Under Varying Textual Contexts

In Sections 3.3 and 3.4, we show that LLMs understand the prompt describing the ego-graph more as a paragraph. Moreover, the benefit of incorporating the structural information is positively related to how homophilous the ego-graph is. Building on these findings, we want to investigate when this benefit might deminish given a fixed set of neighboring nodes. In particular, we are examining the influence of the textual feature richness of the target node. Our study involves experiments on four node classification benchmark datasets with textual node features: CORA (McCallum et al., 2000; Lu & Getoor, 2003; Sen et al., 2008; Yang et al., 2016), PUBMED (Namata et al., 2012; Yang et al., 2016), OGBN-ARXIV (Hu et al., 2020) and OGBN-PRODUCT (Hu et al., 2020)[6].

**Richness of textual node features.** To examine how the richness of the target node's textual features affects text classification accuracy, we compare two different settings:

- *Rich textual context.* In this setting, the nodes are associated with abundant textual features. Specifically, in citation networks (CORA, PUBMED and OGBN-ARXIV), both the paper title and abstract are associated with each node as textual features. In the co-purchasing network (OGBN-PRODUCT), both the product title and product content are associated with each node as textual features. This setting is adopted by several prior studies (Chen et al., 2023; Ye et al., 2023; Guo et al., 2023; Wang et al., 2023; He et al., 2023).

- *Scarce textual context.* In this setting, the nodes are associated with limited textual features. In citation networks (CORA, PUBMED and OGBN-ARXIV), only the paper title is used as textual features. In product networks (OGBN-PRODUCT), only the product name is associated with each node as textual features. While this setting is less explored in the literature, it is of great practical importance due to the prevalence of short texts in social networks (Alsmadi & Gan, 2019). Such limited textual features present challenges like feature sparseness and non-standardization, reducing the effectiveness of traditional methods (Song et al., 2014). In such scenarios, we expect the structural information becomes more useful for the predictions.

**Experimental results.** The experimental results of different prompting methods under the two settings with different richness of textual context are shown in Table 7. We have the following observations: When LLMs are powerful enough, they only benefit from structural information when node feature are scarce: if the node has enough relevant phrases in its own features, the local neighborhood won't be useful anymore. For ChatGPT under rich textual context, the improvement from structural information is minimal. While this improvement is significantly larger when the node level features are scarce.

In conclusion, LLMs only benefit from structural information when the target node does not contain enough phrases for the model to make reasonable prediction.

---

[6]Please see Appendix B.1 for the details of the datasets.

Table 7: Classification accuracy of ChatGPT API for the OGBN-ARXIV, CORA, PUBMED, and OGBN-PRODUCT datasets. ↑ denotes the improvements of best prompt style that leverages structural information over zero-shot method. Best results are **in bold**.

| Textual context | Prompt style | OGBN-ARXIV | CORA | PUBMED | OGBN-PRODUCT |
|---|---|---|---|---|---|
| Rich | Zero-shot | 74.0 | 66.1 | 88.6 | 83.7 |
| | Few-shot | 72.9 | 65.1 | 85.0 | 83.8 |
| | Zero-shot CoT | 71.8 | 56.6 | 81.9 | 80.5 |
| | 1-hop title+label | **75.1** | 72.5 | 89.1 | 85.2 |
| | 2-hop title+label | 74.5 | **74.7** | **89.7** | **86.2** |
| | 1-hop attention | 74.7 | 72.5 | 88.8 | **86.2** |
| | ↑ | 1.1 | 8.6 | 1.1 | 2.5 |
| Scarce | Zero-shot | 69.8 | 61.8 | 85.7 | 78.5 |
| | 1-hop title | 72.3 | 69.6 | 84.8 | 80.5 |
| | 1-hop title+label | **74.3** | 73.9 | 86.4 | 85.3 |
| | 2-hop title | 71.3 | 69.9 | 86.2 | 80.6 |
| | 2-hop title+label | 74.2 | 74.5 | **86.9** | **85.4** |
| | 1-hop attention | 71.3 | **74.7** | 85.1 | 83.9 |
| | ↑ | 4.5 | 12.9 | 1.2 | 6.9 |

## 4 Conclusions and Future Work

In conclusion, our study provides key insights into the application of LLMs in processing structured data, particularly in node classification tasks on text-rich graphs. In this study, we have adapted node classification datasets with textual features from graph learning benchmarks to establish a testbed for LLMs augmented with structured data. By curating a new dataset, we show that data leakage is not the major contributor of LLMs on common node classification benchmarks. Our findings also raise critical concerns regarding the actual depth of understanding that LLMs have of structured data. By two adversarial experiments, we show that LLMs actually do not understand the prompts as graphs structures. On the contrary, LLMs understand the prompts more as paragraphs with augmented keywords. As a consequence, LLMs only show improvement from structural information when the neighborhood is homophilous. Moreover, our findings suggest that when the target node itself contains a wealth of relevant phrases, the additional structural information becomes redundant.

This study also opens several avenues for future research. Firstly, the findings of this study, as well as the new dataset curated by this work, establish a proper benchmark setup for more advanced methods to encode structural information for LLMs, such as finetuning or adapter training. Secondly, while we find that data leakage is not a major concern for the prompting methods examined in this paper, it is still possible that more advanced methods can elicit the memory of the LLMs from training corpus. We may need further investigation on the data leakage issue when proceeding with evaluating other methods. Finally, our results imply that LLMs might be relying on shallow, surface-level patterns rather than grasping the underlying relational complexity of graph structures. Future research may be aimed at developing methods that enable LLMs to deeply parse and comprehend graph topologies.

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

## A   Details about Prompting Format and Settings

We utilize a two-part prompt structure, in line with the ChatGPT Chat Completions API[7] and LLaMA-2-7B format. Each call involves a system prompt and a user prompt. The system prompt, detailed in Table 8, sets LLM's objective and return format. The user prompt, outlined in Table 1, provides information on the target node and its neighborhood for prediction. To standardize LLM's output format, we append "Do not give any reasoning or logic for your answer" to the end of all prompts, except zero-shot CoT prompts.

**Prompt styles.**   Here we introduce the design of prompt styles in our experiments. The exact prompt templates can be found in Table 1.

We first have a few prompt styles that do not encode structural information.

- *Zero-shot*: LLMs make zero-shot predictions based on the target node's textual features only.

- *Few-shot*: LLMs make predictions on nodes' textual features only but with few-shot examples from the training set.

---

[7]`https://platform.openai.com/docs/guides/gpt/chat-completions-api`

- *Zero-shot Chain-of-Thought (CoT)*: Adding "Let's think step by step" to the end of the zero-shot prompt (Kojima et al., 2022). This simple change has been shown to boost LLMs' performance on various tasks comparable to CoT prompts (Wei et al., 2022).

Then we have two strategies for prompt design conceptually inspired by MPNNs, where information from neighboring nodes is aggregated to enhance the representation of the target node:

The first strategy incorporates randomly selected neighbors into the prompt. The idea behind this strategy is to aggregate information from neighboring nodes, following the paradigms of GCN (Kipf & Welling, 2016) and GraphSAGE (Hamilton et al., 2017). The inclusion of 1-hop neighborhood information in the prompt can be seen as an analogous operation to a single-layer aggregation in GCN, where messages from direct neighbors are aggregated. Specifically, we have two styles:

- *k-hop title*: LLMs make predictions based on the target node's textual features as well as titles of neighbors up to k-hop.

- *k-hop title+label*: In addition to *k-hop title*, we include the labels for neighbors in training set or validation set.

The second strategy is designed to weigh the influence of neighboring nodes during the prediction process. This strategy is inspired by Graph Attention Networks (GAT) (Veličković et al., 2017), which employ attention mechanisms to dynamically allocate weights to neighboring nodes based on their task-specific importance. The strategy consists of two steps. a) *Attention extraction*: the LLM ranks neighbors based on their relevance to the target node. b) *Attention prediction*: the LLM makes predictions based on the target node and top-ranked neighbors. We name the whole strategy as *k-hop attention* in our experiment results.

Table 8: System prompts for each dataset.

| Dataset | System Prompt |
|---|---|
| OGBN-ARXIV, ARXIV-2023 | Please predict the most appropriate arXiv Computer Science (CS) sub-category for the paper. The predicted sub-category should be in the format 'cs.XX'. |
| CORA | Please predict the most appropriate category for the paper. Choose from the following categories:\nRule Learning\nNeural Networks\nCase Based\nGenetic Algorithms\nTheory\nReinforcement Learning\nProbabilistic Methods\n |
| PUBMED | Please predict the most likely type of the paper. Your answer should be chosen from:\nType 1 diabetes\nType 2 diabetes\nExperimentally induced diabetes.\n |
| OGBN-PRODUCT | Please predict the most likely category of this product from Amazon. Your answer should be chosen from the list:\nHome & Kitchen\nHealth & Personal Care\n... |

We outline the details for each prompting method as follows:

1. *Few-shot*: Two correct example predictions from ChatGPT are added before the target node information.

2. *Target node with neighbors*: For datasets OGBN-ARXIV, CORA, PUBMED and ARXIV-2023, prompts include up to 20 one-hop and 5 two-hop neighbors. For OGBN-PRODUCT, up to 40 one-hop and 10 two-hop neighbors are included.

3. *Attention extraction*: The maximum number of neighbors is the same as *Target node with neighbors*. We only consider one-hop attention in this study, setting the attention number $k$ to 5.

Common settings for all methods include a temperature of 0 and a maximum output token limit of 500. If a neighbor belongs to the training or validation set, its label is included in the prompt.

# B  Datasets Information

In this section we detail the information about benchmark datasets and the collection pipeline of ARXIV-2023.

## B.1  Datasets Statistics and Splits

Table 9 presents basic statistics for each dataset. For detailed information on datasets and methods to obtain raw text attributes, please see Appendix A in  Chen et al. (2023).

Table 9: Statistics of datasets. Data cut-off indicates the latest data coverage of the dataset.

| Dataset | #Nodes | #Edges | #Task | Metric | #Test Nodes | Data Cut-Off |
|---|---|---|---|---|---|---|
| CORA | 2,708 | 5,429 | 7 | Accuracy | 542 | 2000 |
| PUBMED | 19,717 | 44,338 | 3 | Accuracy | 1,000 | 2000 |
| OGBN-ARXIV | 169,343 | 1,166,243 | 40 | Accuracy | 1,000 | 2019 |
| OGBN-PRODUCT | 2,449,029 | 61,859,140 | 1 | Accuracy | 1,000 | 2019 |
| ARXIV-2023 | 33,868 | 305,672 | 40 | Accuracy | 668 | 2023 |

The dataset splits are as follows:

1. CORA: Training/Validation/Testing ratios are 0.1/0.2/0.2.

2. PUBMED: Training/Validation/Testing ratios are 0.6/0.2/0.2, following  He et al. (2023).

3. OGBN-ARXIV: Original OGB (Hu et al., 2020) splits are used, categorizing papers by their publication year: training (pre-2017), validation (2018), and testing (2019).

4. OGBN-PRODUCT: Original OGB splits are used based on sales ranking: top 8% for training, next 2% for validation, and the remainder for testing.

5. ARXIV-2023: Year-based splits similar to OGBN-ARXIVis adopted: training (pre-2019), validation (2020), and testing (2023).

Due to API cost and rate limits, we test on a random sample of 1,000 nodes for PUBMED, OGBN-ARXIV, and OGBN-PRODUCT, using a fixed seed for reproducibility.

## B.2  Collection of arxiv-2023

The detailed pipeline is as follows:

1. Sample 668 test nodes from around 46,000 arXiv CS papers published from January 1 to August 22, 2023.

2. Extract references to identify one-hop and two-hop neighbors. References were obtained by two steps. First, we search for valid arXiv IDs within each paper, following a method similar to  (Clement et al., 2019). Second, we use AnyStyle to extract the titles of the references,[8] which we then search for via the arXiv API.[9] Titles found on arXiv are considered valid citations if they have a small levenshtein distance (Miller et al., 2009) from the searched title. To prevent duplicate searches, we skip any references that already have a matched arXiv ID. To comply with the arXiv API's rate limit, each paper is restricted to a maximum of 30 searches. For papers published before 2019, we attempt to match them to nodes in the OGBN-ARXIV based on titles. Unmatched pre-2019 nodes are excluded from our dataset.

---

[8]https://github.com/inukshuk/anystyle
[9]https://info.arxiv.org/help/api/basics.html

3. Construct a citation network using nodes from step 2. Basically for each node we need a list of paper it cites. While references for test nodes and one-hop nodes are obtained through both arXiv ID matching and title searching, the references for two-hop nodes are solely determined by arXiv ID matching, due to rate limit constraints. Dataset statistics are in Table 2. We have similar test node degrees between OGBN-ARXIV and ARXIV-2023.

## C MPNNs as Baselines

**Embedding generation.** We adapt the embedding generation pipeline from Hu et al. (2020) to train a skip-gram model (Mikolov et al., 2013) on corpus comprising titles and abstracts from both OGBN-ARXIV and ARXIV-2023. Each paper's 128-dimensional feature vector is then obtained by averaging the word embeddings in its title.

**Hyperparameter tunning.** Baseline models GCN and SAGE are implemented with PyG (Fey & Lenssen, 2019). For hyperparameter tunning, we perform a random search on the following hyperparameter tuning range for every model following Ma et al. (2022):

- Number of layers: $\{2, 3\}$.

- Hidden size: $\{32, 64\}$.

- Learning rate: $\{.001, .005, .01, .1\}$.

- Dropout rate: $\{.2, .4, .6, .8\}$.

- Weight decay: $\{.0001, .001, .01, .1\}$.

Each model is run on 100 random configurations and each random configuration is run for 3 times on OGBN-ARXIV and ARXIV-2023. The max training epoch number is 2000. When training is finished, we use the model with highest average validation accuracy on the dataset for testing.

## D Additional Analysis on the influence of structural information on LLMs.

### D.1 Classification accuracy on LLaMA-2-7B-chat

The results in the main paper are based on `gpt-3.5-turbo-0613`. Here we test the performance of `LLaMA-2-7B-chat`. The results are shown in Table 10. The model gains significant improvement after incorporating structural information in both rich and scarce textual context. The results align with our observation in the paper with ChatGPT that incorporating structural information actually brings performance improvement in both rich and scarce contexts. But a different observation is that the improvement in scarce textual context is not necessarily higher than the improvement in rich textual context. This may be because LLaMA-2-7B is not able to sufficiently leverage the entire text for the prediction in zero-shot prediction. Combining the results of ChatGPT, the conclusion is that, with powerful enough LLM and rich text (e.g. ChatGPT with rich context), the structural information is marginal. But when the text information is scarce or if the LLM cannot fully utilize the text information, structural information can be significantly helpful.

### D.2 The Nuances of When Structural Information Saturates on LLMs and MPNNs.

We compare the performance increase from incorporating structural information for LLMs and MPNNs respectively in Table 11. The average increase from structural data of ChatGPT on 4 datasets is 2.78% (rich context) and 5.44% (scarce context). But the increase from structural data of MPNNs is 6.98% (rich context) and 14.07% (scarce context), which is significantly higher than the gain of LLMs. It means that The benefit of structural information saturates earlier on ChatGPT than MPNNs.
While it's true that structural information is generally more helpful when text is scarce, **quantitatively ChatGPT behaves differently from GNNs**: the benefit of structural information saturates much earlier

Table 10: Classification accuracy for the OGBN-ARXIV, CORA, ARXIV-2023, PUBMED, and OGBN-PRODUCT datasets on LLaMA-2-7B-chat. ↑ denotes the improvements of best prompt style that leverages structural information over zero-shot method. Best results are **in bold**.

| Textual Context | Prompt Style | OGBN-ARXIV | CORA | ARXIV-2023 | PUBMED | OGBN-PRODUCT |
|---|---|---|---|---|---|---|
| Scarce | Zero-shot | 38.8 | 24.5 | 38.2 | 70.1 | 51.7 |
| | 1-hop title | 51.5 | 44.8 | 45.5 | 70.9 | 52.8 |
| | 1-hop title+label | **58.0** | **71.0** | **53.4** | **75.5** | **78.9** |
| | ↑ | 19.2 | 46.5 | 15.2 | 5.4 | 27.2 |
| Rich | Zero-shot | 45.1 | 18.1 | 45.1 | 71.6 | 51.3 |
| | 1-hop title | 51.6 | 51.5 | 50.0 | 68.8 | 52.1 |
| | 1-hop title+label | **66.9** | **66.7** | **60.2** | **73.0** | **77.2** |
| | ↑ | 21.8 | 48.6 | 15.1 | 1.4 | 25.9 |

Table 11: Classification accuracy for the OGBN-ARXIV, CORA, ARXIV-2023, PUBMED on ChatGPT as well as GCN, SAGE and MLP. ↑ (LLMs) denotes the improvements of best prompt style that leverages structural information over zero-shot method. ↑ (MPNNs) denotes the improvements of the best MPNNs over MLP (without structural information).

| Textual Context | Prompt Style | OGBN-ARXIV | CORA | ARXIV-2023 | PUBMED |
|---|---|---|---|---|---|
| Rich | Zero-shot | 74.0 | 66.1 | 73.5 | 88.6 |
| | 1-hop title+label | 75.1 | 72.5 | 73.8 | 89.1 |
| | 2-hop title+label | 74.5 | 74.7 | 73.2 | 89.7 |
| | 1-hop title+label, attention | 74.7 | 72.5 | 73.7 | 88.8 |
| | ↑ (LLMs) | 1.1 | 8.6 | 0.3 | 1.1 |
| | MLP | 69.9 | 65.4 | 69.7 | 86.2 |
| | GCN | 75.4 | 83.0 | 70.3 | 88.4 |
| | SAGE | 75.0 | 83.2 | 70.9 | 90.0 |
| | ↑ (MPNNs) | 5.5 | 17.8 | 1.3 | 3.8 |
| Scarce | Zero-shot | 69.8 | 61.8 | 66.6 | 85.9 |
| | 1-hop title | 72.3 | 69.6 | 70.7 | 80.8 |
| | 1-hop title+label | 74.3 | 73.9 | 70.4 | 84.7 |
| | 2-hop title | 71.3 | 69.9 | 68.9 | 83.5 |
| | 2-hop title+label | 74.2 | 74.5 | 68.5 | 86.4 |
| | ↑ (LLMs) | 4.5 | 12.7 | 4.1 | 0.5 |
| | MLP | 61.9 | 55.7 | 58.5 | 82.0 |
| | GCN | 74.8 | 81.2 | 70.3 | 87.1 |
| | SAGE | 74.4 | 78.8 | 69.1 | 87.9 |
| | ↑ (MPNNs) | 13.0 | 25.6 | 11.8 | 6.0 |

than GNNs with moderate rich textual features; and this is non-trivial since LLaMA-2 doesn't saturate as early as ChatGPT. The average increase from structural data on 4 datasets for ChatGPT/MPNNs/LLaMA-2-7B-chat are 2.78%/6.98%/21.7% respectively.

# E  Additional Analysis for Dropping experiments

**Details about dropping experiments.**  We have three different strategies: a) drop the neighbors with same label (*drop same*), b) drop the neighbors with different label (*drop different*), c) drop neighbors randomly (*drop random*). Let's define $x$ as the number of neighbors with the same ground truth label as the target node, and $y$ as the number of neighbors with a different label from the target node. Given a dropping percentage $p$, we elaborate on the three strategies:

1. *drop random*: We randomly drop $(x + y)p$ neighbors.

2. *drop same*: We retain $\max(x - (x + y)p, 0)$ neighbors with the same labels as the target node while preserving all $y$ neighbors with different labels.

3. *drop different*: We retain $\max(y - (x + y)p, 0)$ neighbors with the different labels from the target node while preserving all $x$ neighbors with same labels.

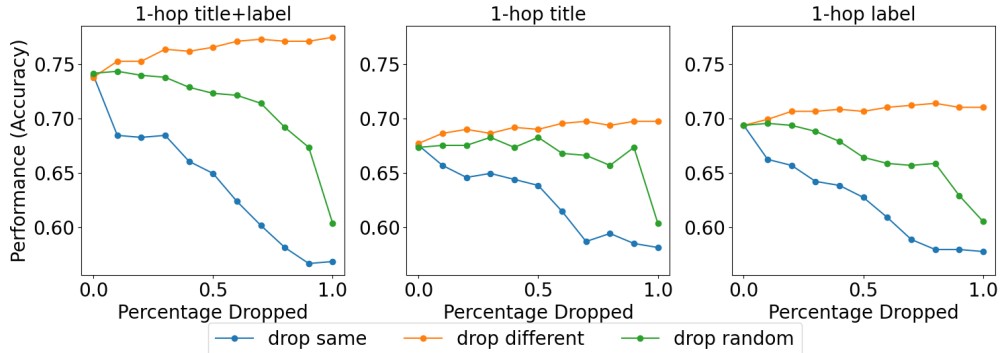

Figure 4: Performance comparison of dropping neighbors using different strategies on CORA dataset. Three dropping strategies are evaluated: (i) 1-hop title+label, (ii) 1-hop title and (iii) 1-hop label

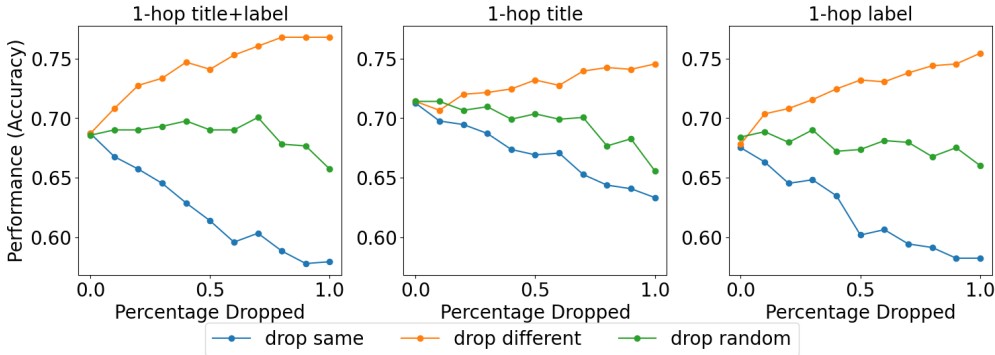

Figure 5: Performance comparison of dropping neighbors using different strategies on ARXIV-2023 dataset. Three dropping strategies are evaluated: (i) 1-hop title+label, (ii) 1-hop title and (iii) 1-hop label

We further explain this by an example. Assume node *A* has 10 neighbors and 6 of the neighbors have same labels as *A*. When dropping percentage is 0.5, *drop same* strategy drops 5 nodes with same label, resulting in 1 neighbor with same label and 4 neighbors with different labels. *drop different* strategy drops all 4 nodes with different labels, resulting in 6 neighbors with same label.

**Ablation study on the effect of labels in the prompt**   We investigate the possibility that LLMs are relying on a simple majority vote in its prediction. We propose a new neighbor dropping experiment with three different prompting styles for neighbors: (i) 1-hop title+label, (ii) 1-hop title and (iii) 1-hop label. 1-hop label means that we only include the label of the neighboring papers, which is used as an ablation study to gauge whether LLM is performing a majority vote based on label information.

If LLMs do rely on a majority vote to determine its prediction. We would expect that the "drop different" curve with 1-hop label goes higher than 1-hop title+label because we are dropping more and more neighbors with different labels. However, we are not observing this in Figure 4 and 5, and the 1-hop label curve is lower than 1-hop title+label curve. This observation refutes the hypothesis that LLMs rely on simple majority vote for prediction. Instead, including more context information will help LLMs to make more accurate predictions as 1-hop title+label "drop different" curve is higher than 1-hop label "drop different" curve.

**Adding Neighbors instead of Dropping.**   To further investigate whether LLMs are benefiting from structural information, we conducted an additional experiment of adding neighbors instead of dropping neighbors. One could argue that adding neighbors that has different groundtruth labels as the target node can still provide some benefits. The results is shown in 6. For all three datasets, adding neighbors with different labels will decrease the prediction performance.

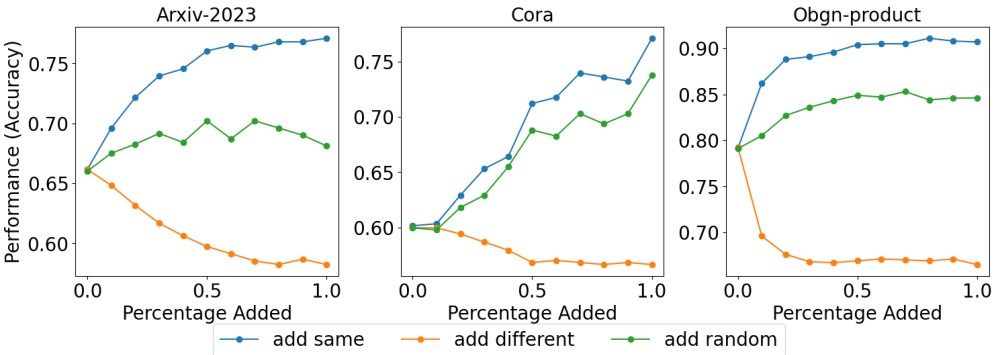

Figure 6: Performance comparison of adding neighbors using different strategies across ARXIV-2023, CORA, and OGBN-PRODUCT datasets. Three adding strategies are evaluated: "add same" adds neighbors with the same label as the target node; "add different" adds neighbors with different labels as the target node; and "add random" randomly selects neighbors for addition. When percentage is 1, "add same" strategy adds all same-label neighbors but excludes all different-label neighbors, and "add different" strategy adds all different-label neighbors but excludes all same-label neighbors.

## F  Additional Analysis for Data Leakage

**Investigating data leakage through prompt variability.**  Chen et al. (2023) reveal considerable fluctuations in Language Model (LLM) performance on OGBN-ARXIVwhen using three distinct prompt words: "arXiv cs subcategory," "arXiv identifier," and natural language. These variations have been interpreted as potential indicators of data leakage.

To delve deeper into this issue, we expand upon their experiments by testing additional prompt words. We also introduce two experimental settings: one with label options provided and another without. As displayed in Table 12, the relative efficacy of various prompts on OGBN-ARXIV mirrors their performance on ARXIV-2023. Importantly, prompts with options underperform on both datasets, underscoring a consistent trend.

Also, utilizing structural information in the prompts can somewhat mitigate the performance drop from less effective prompts. Indicate that LLMs can leverage structural information to improve predictions. This further supports that there is no conclusive evidence for data leakage.

Table 12: Performance across different prompt types between OGBN-ARXIV and ARXIV-2023.

| System Prompt | Zero-shot | | 1-hop title+label | |
|---|---|---|---|---|
| | OGBN-ARXIV | ARXIV-2023 | OGBN-ARXIV | ARXIV-2023 |
| Please predict the most appropriate arXiv Computer Science (CS) sub-category for the paper. The predicted sub-category should be in the format 'cs.XX'. | 74.0 | 73.7 | 74.3 | 70.4 |
| Please predict the most appropriate arXiv Computer Science (CS) sub-category for the paper. Your answer should be chosen from cs.AI, ..cs.SY. The predicted sub-category should be in the format 'cs.XX'. | 66.0 | 68.1 | 70.7 | 67.9 |
| Please predict the most appropriate original arXiv identifier for the paper. The predicted arxiv identifier should be in the format 'arxiv cs.xx'. | 71.3 | 70.8 | 73.7 | 67.5 |
| Please predict the most appropriate original arXiv identifier for the paper. Your answer should be chosen from cs.ai,.. cs.sy. The predicted arxiv identifier should be in the format 'arxiv cs.xx'. | 58.4 | 57.2 | 71.7 | 64.2 |
| Please predict the most appropriate category for the paper. Your answer should be chosen from "Artificial Intelligence",.. "Systems and Control". | 54.6 | 53.4 | 74.1 | 67.8 |

