# OpenReview forum: "Can LLMs Effectively Leverage Graph Structural Information through Prompts, and Why?"
_TMLR — Accepted by TMLR_

### Review · Reviewer_H17Y · 2024-03-19

**Summary Of Contributions:**

This paper studies the ways in which adding graph structural information to a language model's prompt improves prediction performance, and compares LLM models to message-passing neural networks across a number of graph variants. The authors study three aspects of the performance: (1) the impact of possible data leakage in previous benchmarks, (2) the importance of the topological graph information in the LLM prompts, and (3) the impact of graph homophily and of textual node features on the model performance. To study these, the authors introduce a new arxiv-2023 with a low chance of data leakage, and conduct a number of comparisons between MPNNs, ChatGPT, and LLaMA-2. Overall, the authors find that data leakage is unlikely to be a large factor, that LLM models do not strongly depend on the topology within 2-hop neighborhoods, that homophily is correlated with LLM performance, and that LLMs rely less on the graph structure if textual node features are informative.

**Audience:**

Yes

**Broader Impact Concerns:**

No broader impact concerns.

**Claims And Evidence:**

Yes

**Requested Changes:**

Before recommending this paper for acceptance, I think the authors should:

- Modify the claims about the experiments in section 3.3 to reflect the type of prompts used by previous work.
- Clarify how the edge information in the "random" and "extreme" settings is actually being used in the prompts, if it is at all, and why it would be expected to have an impact. (If the prompts are exactly the same for these settings, I think the experiment doesn't really show anything useful.)
- Add details about when and how the ChatGPT data was collected, and whether there is any possibility of data leakage due to model updates after ChatGPT's initial launch.
- Either reword the claims about homophily to make them more speculative or more precise, or include some extra experiments to show that including non-homophilous nodes is actually worse than including no structural information at all.

Beyond this, I think there are some other ways the paper could be improved as well:

- I think the homophily section would benefit from having more discussion of the "complicated mechanism" by which more homophilous nodes are easier to classify. Is there any hypothesis about what this mechanism is, or any other property that these nodes share?
- The conclusion states that the results "imply that LLMs might be relying on shallow, surface-level patterns rather than grasping the underlying relational complexity of graph structures". This seems to make a strong assumption that the LLMs are missing some deeper structure. Perhaps the text information already contains enough information to infer the graph structure, so the model is already using the graph structure even if it isn't being provided? Or perhaps the graph structure is actually not very useful for this task. Given that the LLM is matching or outperforming the MPNNs in the 1-hop title+label setting, it seems possible that the LLMs are already using the graph structure as much as they need to. I think this possibility could be discussed more.
- I found Table 1 to be very difficult to understand, and it seems somewhat incomplete. For instance, what does "It is linked to paper <list of neighbor papers>... (more 1-hop neighbors)" mean? Does each paper in the list have it's own "It is linked to paper" statement or is it just a list of paper titles? Is paper 1 necessarily  a 1-hop neighbor? Also, for "It has following neighbor papers at hop 2:", are you including any actual edges in the prompt? Can the LLM tell anything about the topology beyond the set of 1-hop and 2-hop neighbors?
- Some typos: "rigorours" on line 85, "introduces=" on line 95.

**Strengths And Weaknesses:**

## Strengths

### [S1] New graph dataset of recent papers, avoiding data leakage

The authors provide a new dataset, arxiv-2023, designed to control for data leakage in language models. This dataset follows the construction of the obgn-arxiv dataset as closely as possible, but is restricted to recent CS papers, making it a useful dataset for comparing models that potentially trained on papers in the obgn-arxiv dataset.

### [S2] Thorough experimental results

The paper includes many experimental results studying multiple aspects of model performance. The authors carefully compare results for multiple prompting strategies and for two baseline MPNNs, between obgn-arxiv and arxiv-2023, between different graph modifications, across different amounts of homophily, and across different amounts of text content.

### [S3] Interesting analyses of homophily and amount of text context

The results about homophily and text content are interesting. The authors find that dropping neighbors to increase or decrease homophily leads to an increase or decrease in predictive accuracy, showing that LLMs are using homophily in their classification. They also somewhat surprisingly find that homophilous nodes are easier to classify even without including the neighbors, indicating some sort of correlation between homophily and ease of classification. For the impact of text context, the authors find that models improve more by including structural information when there is less textual context available in the node features.

## Weaknesses

### [W1] Misleading claims and confusing experiment design in Sec. 3.3 ("treating the prompts as graphs")
*Edit: Addressed by the authors in the revised submission.*

Section 3.3 is claimed to demonstrate that "LLMs interpret inputs more as contextual paragraphs than as graphs with topological structures, which contradicts the intention of prompt designers". Based on the experiments the authors describe, this seems misleading and not fully correct.

It's not clear to me that the cited previous works on this topic expect the LLMs to fully understand the graph topology, so it's not clear that the findings here are really a contradiction. As far as I can tell, all three of the papers the authors cite that encode the ego-graph as prompts (Chen et al., 2023; Guo et al., 2023; Hu et al., 2023) are already using a linearized representation. So it seems that those prompt designers are already intentionally treating the inputs as "contextual paragraphs" and aren't relying on the ability of the model to pick out specific edges between neighbors.

Also, from my understanding of the "k-hop title" format, the prompt template doesn't include the edges between the 2-hop neighbors and the 1-hop neighbors at all? So it seems like reconnecting 2-hop neighbors to different 1-hop neighbors (e.g. in the "random" or "extreme" settings) would not affect the prompt. If you aren't even telling the model about those edges in the first place, isn't it obvious that the model performance would stay the same when you change those edges? (It's also possible I'm misunderstanding the prompt format; I found Table 1 very hard to understand.)

And even so, it's not clear to me why doing this rewiring would significantly change performance even if the model was fully capable of using graph information. Even the "extreme" setting seems to preserve the relative distances between the ego-node and each other paper. The exact connections between the 1-hop neighbors and the 2-hop neighbors seem relatively much less important, and I don't see why a model would benefit from paying attention to these connections.

Overall it seems like the takeaway here is that the LLMs get good performance even without edge information? But this seems like a different claim than saying LLMs *don't* interpret the inputs as graphs with topological structures. Perhaps they are fully capable of using that information, but it's either missing or not actually useful in this task, so they don't use it.

### [W2] Possibility of data leakage in ChatGPT experiment
*Edit: Addressed by the authors in the revised submission.*

The authors state that ChatGPT's training cutoff is September 2021. However, the current training cutoff for ChatGPT appears to be at least April 2023 (see https://chat.openai.com/share/3b088460-86f8-4fa6-afd1-802b7f9a5e9f). I couldn't find any discussion of this in the paper. When did you query ChatGPT, and did you make sure that its reported training cutoff was indeed prior to the cutoff date of your arxiv-2023 dataset?

(Note that this doesn't affect the LLaMA-2 results, so I don't think it significantly weakens the paper. Also, if there was a large amount of data leakage in obgn-arxiv, I would have expected it to be weaker for the 2023 data, even if the time windows overlap a bit. But it seems important to be precise here about how sure you are that there is not any data leakage.)

### [W3] Somewhat overly-broad claims about homophily
*Edit: Addressed by the authors in the revised submission.*

Although I think the homophily experiment is interesting, I'm not convinced by all the claims about it. The authors state "LLMs benefit from structural information only when the neighborhood is homophilous, meaning the neighbors contain phrases related to the groundtruth label of the target node" and "LLMs only show improvement from structural information when the neighborhood is homophilous". I don't think the experiment provides direct evidence for this? The experiment shows that removing nodes with the correct label reduces performance, but I don't think it shows that the LLM is not benefitting at all from graph structure, relative to the zero-shot setting. Perhaps they are still providing some benefit, but it's just a much smaller benefit than the ones with correct labels.

I think properly supporting this claim would require comparing the performance between having no structural information and having non-homophilous structural information. If you start from zero neighbors, and you selectively add non-homophilous nodes to the context, does performance actually go down, or does it just increase more slowly?

---

> ### Author Response · Authors · 2024-04-15
> **Response to Reviewer H17Y part 1**
>
> Thank you for your comments!
>
> > [Weakness 1.1] It's not clear to me that the cited previous works on this topic expect the LLMs to fully understand the graph topology, so it's not clear that the findings here are really a contradiction.
>
> Thanks for pointing this out, below we clarify how related literature encodes structural information into prompts.
>
> First, we clarify that the **linearized prompt in our context is free of any kind of structural information**. We basically concatenate all the titles and abstracts into one single paragraph, without guiding LLMs which paper is the paper to be predicted.
>
> Here is an example of our linearized prompt:
> ```
> Abstract: <abstract>\nTitle: <title>
> <neighbor paper 1 title>\nLabel: <neighbor paper 1 label>\n
> <neighbor paper 2 title>\nLabel: <neighbor paper 2 label>\n
> (more neighbors)\nDo not give any reasoning or logic for your answer. \nAnswer: \n\n
> ```
>
> Compared with existing literature:
> - Chen et al., 2023 uses two prompts styles with listing all neighbors (Table 16 in their paper). For example:
> ```
> Instruction: <Task instruction>
> Paper 1: <paper 1 content>
> Paper 2: <paper 2 content> ...
> Paper 1 cites Paper 2, Paper 1 cites Paper 3....
> ```
> - Hu et al., 2023 use a list of neighbors’ titles to encode the structural information (Table 5 in their paper).
>
> - Guo et al., 2023 use a neighbor summary and distinguish between the target paper and the summary. But it’s not clear whether they have used structural information when generating the summary.
>
> - Tang et al. designs a template with explicit edge index (Figure 4 in their paper).
> ```
> Graph Information: <graph>: Central Node: 2, Edge index: [[...src node...],[...dst node...]], Node list: [...]
> ```
>
> - Das et al. uses a template with explicit adjacency list for node classification (Figure 2 in their paper).
> ```
> Adjacency list :
> {A:[B,C,D], B:[C,D], C:[D,E,F],... K:[A,D]}
> Node-Label Mapping:
> {A: ?, B: 1, C: 1, D:2 ...}
> ```
>
>
> Overall, several related literature includes explicit graph structures, which motivates us to ask whether LLMs actually understand the prompt as intended. We have updated the paper at line  189-193 to reflect the change.
>
>
>
> Chen, Zhikai, et al. "Exploring the potential of large language models (llms) in learning on graphs." ACM SIGKDD Explorations Newsletter 25.2 (2024): 42-61.
>
> Tang, Jiabin, et al. "Graphgpt: Graph instruction tuning for large language models." arXiv preprint arXiv:2310.13023 (2023).
>
> Das, Debarati, et al. "Which Modality should I use--Text, Motif, or Image?: Understanding Graphs with Large Language Models." arXiv preprint arXiv:2311.09862 (2023).
>
> Jiayan Guo, Lun Du, and Hengyu Liu. Gpt4graph: Can large language models understand graph structured data? an empirical evaluation and benchmarking. arXiv preprint arXiv:2305.15066, 2023.
>
> Yuntong Hu, Zheng Zhang, and Liang Zhao. Beyond text: A deep dive into large language models’ ability on understanding graph data. arXiv preprint arXiv:2310.04944, 2023

---

> > ### Author Response · Authors · 2024-04-15
> > **Response to Reviewer H17Y part 2**
> >
> > >  [Weakness 1.2] from my understanding of the "k-hop title" format, the prompt template doesn't include the edges between the 2-hop neighbors and the 1-hop neighbors at all? So it seems like reconnecting 2-hop neighbors to different 1-hop neighbors (e.g. in the "random" or "extreme" settings) would not affect the prompt.
> >
> > The edge information between neighbors **is included** in this part of the k-hop title prompt: “It is linked to paper <list of neighbor papers>”. Below is an example:
> >
> > ```
> > Abstract: <abstract>\nTitle: <title>
> > It has following neighbor papers at hop 1:
> > Paper 1 title: <paper 1 title>\nLabel: <paper 1 label>\nIt is linked to paper 2, 3.
> > Paper 2 title: <paper 2 title>\nLabel: <paper 2 label>\nIt is linked to paper 1, 4.
> > (more neighbors)\
> > Do not give any reasoning or logic for your answer. \nAnswer: \n\n
> > ```
> >
> > In this format, rewiring 2-hop neighbors will lead to different prompts. We have reformatted Table 1 to clarify this.
> >
> >
> >
> > > [Weakness 1.3] Overall it seems like the takeaway here is that the LLMs get good performance even without edge information? But this seems like a different claim than saying LLMs don't interpret the inputs as graphs with topological structures. Perhaps they are fully capable of using that information, but it's either missing or not actually useful in this task, so they don't use it.
> >
> > Thank you for your comments. The take-away here can be interpreted as the explicit description of graph structure does not help the prediction of node labels **on text-attributed graphs**. We have added the “text-attributed graphs”  into the title of Sec 3.3 and paper title to clarify the scope of our claim.
> >
> >
> >
> > > [Weakness 2] Possibility of data leakage in ChatGPT experiment. The authors state that ChatGPT's training cutoff is September 2021. However, the current training cutoff for ChatGPT appears to be at least April 2023.
> >
> > The model used throughout the paper is `gpt-3.5-turbo-0613`, which has a data cutoff up to Sep 2021 (https://platform.openai.com/docs/models/gpt-3-5-turbo). Originally this is stated in the footnote on page 5. To make this more clear for readers, we have further clarified this in the revised paper (line 139).
> >
> >
> >
> > > [Weakness 3] Somewhat overly-broad claims about homophily. I think properly supporting this claim would require comparing the performance between having no structural information and having non-homophilous structural information. If you start from zero neighbors, and you selectively add non-homophilous nodes to the context, does performance actually go down, or does it just increase more slowly?
> >
> > Thank you for providing this perspective. We have conducted experiments as required. **We include a new experiment to add neighbors instead of dropping neighbors, starting from zero neighbors. The result is included at Page 20, Figure 6.**
> > In all three datasets, adding non-homophilous nodes will decrease the prediction accuracy. This experiment further supports our claim about the positive relationship between local homophily ratio and prediction correctness.

---

> > > ### Author Response · Authors · 2024-04-15
> > > **Response to Reviewer H17Y part 3**
> > >
> > > ## Requested Changes
> > >
> > > - > “Modify the claims about the experiments in section 3.3 to reflect the type of prompts used by previous work.”
> > >   - Answered in weakness 1.1.
> > > - > “Clarify how the edge information in the "random" and "extreme" settings is actually being used in the prompts, if it is at all, and why it would be expected to have an impact.”
> > >   - Answered in weakness 1.2.
> > > - > “Add details about when and how the ChatGPT data was collected”
> > >   - Answered in weakness 2.
> > > - > “Either reword the claims about homophily to make them more speculative or more precise, or include some extra experiments to show that including non-homophilous nodes is actually worse than including no structural information at all.”
> > >   - Answered in weakness 3.
> > >
> > > - > “I think the homophily section would benefit from having more discussion of the "complicated mechanism" by which more homophilous nodes are easier to classify. Is there any hypothesis about what this mechanism is, or any other property that these nodes share?”
> > >   - One possible explanation for the observation that homophilous nodes are easier to be classified is that, nodes that are less homophilous are more likely to be multidisciplinary. This will make less homophilous nodes harder to predict in the first place. We added the discussion in line 276-278 in the revised paper.
> > > - > “The conclusion states that the results "imply that LLMs might be relying on shallow, surface-level patterns rather than grasping the underlying relational complexity of graph structures". This seems to make a strong assumption that the LLMs are missing some deeper structure. Perhaps the text information already contains enough information to infer the graph structure, so the model is already using the graph structure even if it isn't being provided?”
> > >   - Our experiments in Sec 3.5 actually suggest that text attributes are the most important component for LLMs to predict the node labels, instead of structural information. For example, comparing the two settings with different richness of textual context (Table 7), we show that if the node has enough relevant phrases in its own features, the local neighborhood won’t be useful anymore.
> > > - > “I found Table 1 to be very difficult to understand, and it seems somewhat incomplete.”
> > >   - > ”For instance, what does "It is linked to paper <list of neighbor papers>... (more 1-hop neighbors)" mean? Does each paper in the list have it's own "It is linked to paper" statement or is it just a list of paper titles?”
> > >     - Each neighbor paper has its own list of linked papers. We have reformatted Table 1 to clarify this.
> > >   - > “Is paper 1 necessarily a 1-hop neighbor?”
> > >     - In k-hop title prompting style, all the papers at hop k will be led by the sentence “It has following neighbor papers at hop k:”. When listing neighbors, we start from 1-hop neighbors and extend to more distant neighbors. So usually paper 1 in the prompt is a 1-hop neighbor.
> > >   - > “Can the LLM tell anything about the topology beyond the set of 1-hop and 2-hop neighbors?”
> > >     - Yes, See the example below.
> > >
> > > ```
> > > Abstract: <abstract>\nTitle: <title>
> > > It has following neighbor papers at hop 1:
> > > Paper 1 title: <paper 1 title>\nLabel: <paper 1 label>\nIt is linked to paper 2, 5.
> > > Paper 2 title: <paper 2 title>\nLabel: <paper 2 label>\nIt is linked to paper 1, 6
> > > (more 1-hop neighbors..)
> > > It has following neighbor papers at hop 2:
> > > Paper 5 title: <paper 5 title>\nLabel: <paper 5 label>\nIt is linked to paper 1, 6.
> > > Paper 6 title: <paper 6 title>\nLabel: <paper 6 label>\nIt is linked to paper 2, 5.
> > > (more 2-hop neighbors..)
> > > ```
> > >
> > > - > “Some typos: "rigorours" on line 85, "introduces=" on line 95.”
> > >   - Thanks for pointing out, we have fixed the typos in the revised paper.

---

> > > > ### Author Response · Authors · 2024-04-18
> > > > **Response to Reviewer H17Y**
> > > >
> > > > Dear Reviewer H17Y, thank you again for your effort in reviewing our paper. As the end of the discussion period approaches, we would like to ask if our responses were able to sufficiently address your concerns. If you have further questions, please let us know and we are eager to further address them!

---

> ### Comment · Reviewer_H17Y · 2024-04-20
> **Discussion**
>
> Thank you for your reply. My responses are below.
>
> ## Weakness 1.1
>
> ### (a) Chen et al. 2023
> > Chen et al., 2023 uses two prompts styles with listing all neighbors (Table 16 in their paper).
>
> Hm, Table 16 of Chen et al. (2023) (via [this arxiv URL](https://arxiv.org/pdf/2307.03393.pdf)) appears to be about pseudo-labels for GCNs. Did you mean a different table? I don't see any table listing the prompt style.
>
> Also, I see in their paper that they say:
>
> > Using Cora dataset as an example, we try to use prompts like “node 1: ⟨paper content⟩” to represent attributes, and prompts like “node 1 cites node 2” to represent the edge. However, we find that this approach is not feasible since LLMs usually present a small input context length restriction.
>
> I interpret this as saying that the authors *don't* actually use this format. Furthermore, I can't find the word "cites" anywhere in the code provided by Chen et al. Are you sure this is the prompt format they are using?
>
> ### (b) Hu et al., 2023
> It looks like they just use one-hop neighbors. They also use a lot of additional prompt engineering. So it doesn't seem to me they are assigning much weight to the graph topology itself, and your rewirings don't seem very relevant to their format. I don't think Hu et al. would necessarily expect the LLM to "treat the input prompt as a graph" in the way that you mean.
>
> ### (c) Guo et al., 2023
> I agree it is ambiguous exactly how they construct the summary. Given that, it seems unclear that they would expect the input to be treated like a graph in the way you claim.
>
> ### (d) Tang et al. and Das et al.
> Thanks for these new citations, which I agree do use graph structure more explicitly. However, it does seem like the prompts in those works are more sophisticated than the prompts in your Table 1. So it could be that LLMs actually can use the graph information with their prompts, even if they don't with yours.
>
> ### Summary of my concern
> I understand that many previous works have integrated graph structures into prompts in a variety of ways, and it's fine yours doesn't match theirs exactly. But your paper currently seems to claim that
>
> 1. the prompt designers from these previous works expected the LLM to "treat the input prompt as a graph" in a particular sense (specifically, that it depends on the topology in a way that applying your ablations would break it)
> 2. your results indicate that they don't do this, which "contradicts the intention of prompt designers."
>
> My concern is that the prompt designers for the previous papers might not actually see your work as contradicting their expectations. Some of the authors might be using "structural information" just to mean "neighborhood relationships", and others seem to be using more complex prompting strategies (meaning that your prompts might not be a good proxy for them). Since we don't actually know what expectations they had, I think it's too strong of a claim to say that your results contradict their expectations.
>
> To address this concern, I'd be satisfied if you rephrased Section 1 and Section 3.3 to make it clear that you aren't necessarily contradicting the expectations of the authors / prompt designers in this previous work (since it's not stated exactly what these prompt designers expected). Instead, there's a specific *interpretation* of these previous findings that you believe prompt designers might have, and your results contradict this interpretation. It would also be useful for you to specifically state which expectation your results contradict.
>
> ## Weakness 1.2
> > The edge information between neighbors is included in this part of the k-hop title prompt ... We have reformatted Table 1 to clarify this.
>
> Thank you, this was not clear from the original paper but it is clear in the revised Table 1. This mostly addresses my concern.
>
> I'm still not sure why someone would expect moving the two-hop neighbors around to be relevant to this task, though. Why would a LLM need to care about the exact two-hop relationships? As I said in my initial review, "The exact connections between the 1-hop neighbors and the 2-hop neighbors seem relatively much less important, and I don't see why a model would benefit from paying attention to these connections." I think this is relevant in context of Weakness 1.1, since it seems like you think the prompt designers would expect the LLM to care about 2-hop neighbor edges and be surprised it doesn't, but I don't know why someone would have that expectation.
>
> ## Weakness 2
> Thanks for clarifying the specific version of ChatGPT you used. This resolves my concern.
>
> ## Weakness 3
> Thanks for adding the additional results, which do seem to support your claims. Please add a brief mention of these results somewhere in Section 3.4 (possibly at the end of "The neighbor dropping experiment") so that readers know to look for them.
>
> ## Other questions
> Thanks for your responses to my other questions, and for adding the additional discussion of homophilous nodes.

---

> > ### Author Response · Authors · 2024-04-21
> > **Further response to Reviewer H17Y**
> >
> > Thank you for your reply and further comments! We respond below:
> >
> > > Table 16 of Chen et al. (2023) (via this arxiv URL) appears to be about pseudo-labels for GCNs.  Did you mean a different table? I don't see any table listing the prompt style.
> >
> > The Table 16 we are referring to is in [version 3](https://arxiv.org/pdf/2307.03393v3.pdf) of their paper (Page 17). The URL you are referring to is version 4. It turns out that the authors delete the table in the newest version. We interpret that the authors explored the prompting style with explicit graph structure before.
> >
> > > My concern is that the prompt designers for the previous papers might not actually see your work as contradicting their expectations. Some of the authors might be using "structural information" just to mean "neighborhood relationships", and others seem to be using more complex prompting strategies (meaning that your prompts might not be a good proxy for them). Since we don't actually know what expectations they had, I think it's too strong of a claim to say that your results contradict their expectations. To address this concern, I'd be satisfied if you rephrased Section 1 and Section 3.3 to make it clear that you aren't necessarily contradicting the expectations of the authors / prompt designers in this previous work (since it's not stated exactly what these prompt designers expected).
> >
> > Thank you for your comment. We agree that it may be too strong to claim that all previous paper’s expectation on LLMs is to understand explicit graph structures on text-attributed graphs. We have rephrased Sec. 1 and Sec. 3.3 to change the narrative in line 10, 37, 59, 124, 190-191, 322:
> >
> > In the new version, we do not claim that we contradict the expectations of the prompt designers in previous works. Instead, we claim that “It is implied that some previous works hold that LLMs may benefit from describing the explicit graph structures”, but our study shows it’s not the case.
> >
> > We also note that this modification does not change the take-away for Sec 3.3: LLMs interpret inputs prompts more as contextual paragraphs than as graphs with topological structures.
> >
> >
> >
> > > Please add a brief mention of these results somewhere in Section 3.4 (possibly at the end of "The neighbor dropping experiment") so that readers know to look for them.
> >
> > Thanks for the suggestion. It’s added at line 252-253.
> >
> > > > The edge information between neighbors is included in this part of the k-hop title prompt ... We have reformatted Table 1 to clarify this.
> > >
> > > Thank you, this was not clear from the original paper but it is clear in the revised Table 1. This mostly addresses my concern. I'm still not sure why someone would expect moving the two-hop neighbors around to be relevant to this task, though. Why would a LLM need to care about the exact two-hop relationships? As I said in my initial review, "The exact connections between the 1-hop neighbors and the 2-hop neighbors seem relatively much less important, and I don't see why a model would benefit from paying attention to these connections." I think this is relevant in context of Weakness 1.1, since it seems like you think the prompt designers would expect the LLM to care about 2-hop neighbor edges and be surprised it doesn't, but I don't know why someone would have that expectation.
> >
> > Thanks for acknowledging the reformatted Table 1. **We are not just manipulating connections between 2-hop neighbors.** In Sec. 3.3, we introduce two experiments (1) Linearize ego-graph and (2) Rewire ego-graph, to argue that LLMs actually understand the input prompt as a paragraph of text with augmented keywords, instead of a graph. We elaborate below:
> >
> > - In (1) Linearize ego-graph, we create a  linearized version of the graph-structured prompts by only keeping all neighbors’ text attributes in the prompts, which means **LLMs could not tell between all 1-hop and 2-hop neighbors**. Even in this case, **it does not significantly affect the prediction performance**. The average difference between Graph-structured prompt (e.g. 1/2-hop title+label) and Linearized prompt is only 0.7%  (Table 4). It’s surprising that totally ignoring structural information can hardly change the prediction accuracy.
> > - In (2) Rewire ego-graph, we include a “Path” rewiring to randomly connect all 1-hop neighbors into a path (Table 5). In this case, all 2-hop neighbors are removed and the ego-graph structure is significantly altered. However, even this level of rewiring does not significantly change the performance of LLMs (average drop of LLMs is 1.9%, compared to 8.9% drop for MPNNs, for ChatGPT and LLaMA-2-7B on 3 datasets).
> > - Moreover, even just rewiring the 2-hop neighbors will decrease the performance of MPNNs significantly by 3.9% on average, as shown for “Random” and “Extreme” rewiring. So it’s not clear whether LLMs will suffer from the rewiring on the same level as MPNNs.

---

> > > ### Comment · Reviewer_H17Y · 2024-04-23
> > > **Updated review**
> > >
> > > Thanks for updating the paper and for your additional clarifications. I believe the revised submission meets the TMLR acceptance criteria, so I have updated my review accordingly.

---

### Review · Reviewer_UYwb · 2024-03-23

**Summary Of Contributions:**

The paper explores the capability of Large Language Models (LLMs) to utilize structural information of graph data to improve prediction performance, particularly in node classification tasks on text-rich graphs.  Firstly, the authors create a new dataset to address concerns about data leakage potentially inflating LLM performance.  They conclude that LLMs interpret prompts more as linearized paragraphs than graph structures, and argue that LLMs do not benefit from the topological structure of the ego-graph.

**Audience:**

Yes

**Claims And Evidence:**

No

**Requested Changes:**

- Expand the experimental setup to include multiple datasets with varying characteristics and complexities, such as purely structural graph datasets without text information and larger graphs.
- Conduct focused analyses to explicitly explore the role of homophily in the performance of LLMs on graph-structured data. This could involve controlled experiments where the degree of homophily within the graph data is varied.

Overall, I think the paper makes some interesting claims and studies important problems such as data leakage in LLMS, and the impact of graph structure in LLMs. However, it seems to approach these topics at a surface level, lacking in-depth exploration and analysis in both the theoretical and experimental parts.

**Strengths And Weaknesses:**

## Strengths

- The use of adversarial experiments to probe LLMs' understanding of graph structures offers insights into how LLMs process structured information, contributing to the broader understanding of LLM capabilities and limitations in this topic.

- Another strength of this paper is its investigation of the potential impact of data leakage on the performance of LLMs in processing graph-structured data for node classification tasks.

## Weaknesses

- The organization could be improved to better guide the reader through the research questions, methodologies, findings, and implications of the study. It seems that the authors investigated many topics (data leakage, graph structure impact in LLMs.), without going in-depth into any of those.

- The reliance on a single dataset to derive conclusions about the impact of data leakage and the processing of structural information by LLMs limits the generalizability of the findings. Additional experiments with diverse datasets are necessary to strengthen the claims.

- The paper lacks a solid theoretical framework to explain the observed phenomena, especially concerning the processing of graph structures and the role of homophily. The absence of a theoretical rationale makes it difficult to understand the underlying mechanisms at play. Moreover, the paper also doesn't provide strong experimental results.  This combination of theoretical and empirical shortcomings significantly limits the paper's contribution.

---

> ### Author Response · Authors · 2024-04-15
> **Response to Reviewer UYwb part 1**
>
> > The organization could be improved to better guide the reader through the research questions, methodologies, findings, and implications of the study. It seems that the authors investigated many topics (data leakage, graph structure impact in LLMs.), without going in-depth into any of those.
>
> Thanks for the comment. We briefly restate the three research questions and corresponding sections for this paper:
> 1. How does data leakage potentially affect the performance of LLMs on node classification tasks? This is discussed in Sec 3.2.
> 2. In what way do LLMs comprehend structural information, and does this align with the prompt designers' intentions? This is discussed in Sec 3.3.
> 3. What factors contribute to LLMs benefiting from structural information in their performance? This is discussed in Sec 3.4 and 3.5.
>
> We introduce the methodology and findings in each section. If you find any part of the sections not clear, please kindly let us know and we will revise. Thank you!
>
> > "reliance on a single dataset to derive conclusions about the impact of data leakage and the processing of structural information by LLMs limits the generalizability of the findings. Additional experiments with diverse datasets are necessary to strengthen the claims."
>
> For the data leakage part, **although arxiv-2023 is the only leakage-free dataset we collected, We did have concrete evidence for the negligible effect of potential data leakage for the prompting styles that we tried.** If LLMs rely on data leakage to make accurate prediction, the performance drop of LLMs between OGBN-ARXIV (may have leakage problem) and ARXIV-2023 (leakage-free) should **be significantly greater** than the drop on MPNNs on two datasets. This is because LLMs may benefit from their memory on OGBN-ARXIV from data leakage to have higher accuracy. But this advantage is not likely on ARXIV-2023. However, in Table 4, we **observe the contrary**: the performance drop of LLMs between OGBN-ARXIV and ARXIV-2023 **is less than** the drop on MPNNs on two datasets (1.3% compared to 5.1% in rich context, 3.6% vs 4.5% in scarce context). This means that LLMs actually generalize well to leakage-free data.
>
> For all other experiments, we test at least three datasets.
> - Adversarial settings (Sec. 3.3) and homophily (Sec. 3.4): we test over arxiv-2023, ogbn-arxiv and ogbn-product.
> - Influence of varying textual contexts (Sec 3.5): we test over ogbn-arxiv, cora, pubmed and ogbn-product.
>
>
> > The paper lacks a solid theoretical framework to explain the observed phenomena, especially concerning the processing of graph structures and the role of homophily.
>
> Thanks for pointing this out, below we explain more on why we connect graph structure and homophily.
> 1. Homophily (McPherson et al. in 2001), which describes the tendency of nodes to form connections with similar nodes, significantly impacting the success of different graph learning techniques (line 112-121).
> 2. Recent studies also find that the degree of homophily in a dataset often correlates with the efficacy of MPNNs in classification tasks (line 235-238).
>
> The two points above motivates us to ask the question: what is the impact of homophily on the classification performance of LLMs?
>
>
> > Expand the experimental setup to include multiple datasets with varying characteristics and complexities, such as purely structural graph datasets without text information and larger graphs.
>
> The paper focuses on text-attributed graphs, and is motivated by recent literature on combining LLMs and text-attributed graphs. Purely structural graphs are out of the scope of this paper. However, we note that the four existing datasets and one newly-collected dataset also provide different characteristics in terms of dataset domain, graph statistics and difficulties.  We have added the “text-attributed graphs” into the title to clarify the scope of this paper: “Can LLMs Effectively Leverage Graph Structural Information through Prompts **in Text-attributed Graphs**, and Why?”

---

> > ### Author Response · Authors · 2024-04-15
> > **Response to Reviewer UYwb part 2**
> >
> > > Conduct focused analyses to explicitly explore the role of homophily in the performance of LLMs on graph-structured data. This could involve controlled experiments where the degree of homophily within the graph data is varied.
> >
> > Actually in Sec 3.4 and Appendix E we have several experiments to measure the influence of different levels of homophily on prediction accuracy. Below are the experiments and take-aways:
> > - A neighbor dropping experiments to manipulate the level of the local homophily ratio (Page 8, Figure 3). → We argue that LLMs benefit from structural information only when the neighborhood is homophilous.
> > - A correlation study between local homophily ratio and prediction correctness (Page 9, Table 6). → We find a positive relationship between local homophily ratio and prediction correctness.
> > - An ablation study on the effect of labels in the prompt. This is to investigate the possibility that LLMs are relying on a simple majority vote in its prediction (Page 18, Figure 4, 5). → LLMs are not relying on a majority vote to make predictions.
> > - A newly added experiment on adding neighbors instead of dropping neighbors. This is discussed in Appendix E (Page 20, Figure 6). → Adding neighbors with different labels will decrease the prediction performance.

---

> > > ### Author Response · Authors · 2024-04-18
> > > **Response to Reviewer UYwb**
> > >
> > > Dear Reviewer UYwb, thank you again for your effort in reviewing our paper. As the end of the discussion period approaches, we would like to ask if our responses were able to sufficiently address your concerns. If you have further questions, please let us know and we are eager to further address them!

---

### Review · Reviewer_5tBU · 2024-04-07

**Summary Of Contributions:**

This work aims at understanding whether LLMs can leverage graph structural information in some prediction tasks on text-rich graph, such as node label.

To do so, the authors try to understand:

- The role of data leakage and memorization through experiments on a newer version of ogbn-arxiv. I think this question is important for obvious reasons, and it makes sense, instead of trying to measure contamination, which is rather difficult, to just create “unseen” tokens.
- Wether the LLM leverages structural information in the task prompt, mostly by comparing performance with or without graph structure information, or modifying it.
- The importance of graph homophily in LLMs prediction.

For all these sub questions, a clear answer is provided. The conclusion is that LLMs do not really leverage this information for the task of study (text-rich graphs).

**Audience:**

Yes

**Claims And Evidence:**

Yes

**Requested Changes:**

See above: the paper is called "Can LLMs Effectively Leverage Graph Structural Information through Prompts, and Why?". To have an answer to this, it feels the authors should either:
- Do follow-up experiments as to why graph structure seems not useful to LLMs for the task (better prompts? LLMs limitations? Graph info useless to LLMs for this task since text-rich?). Without this, the title of the paper should rather be something like "LLMs do not leverage graph structural information through simple prompts for text-rich tasks".
- Or, bring some clarifications as to why focusing on this particular task and setup.

**Strengths And Weaknesses:**

Strength:

- Studying the ability of LLMs to understand graphs is an important question as it is not clear as of today to what extent text-only training provides LLMs with a sense of physical and spatial information, graph understanding falling in the second category.
- The writing of the paper is mostly clear.
- For each question, the described protocol seems sound which makes me confident about the paper’s conclusions.

Weakness:

It seems to me that this work is in a sense incomplete: the title is "Can LLMs Effectively Leverage Graph Structural Information through Prompts, and Why?", but the conclusion is that LLMs do not seem to leverage this information for the particular task of study.

Hence, it is not clear to me whether this is because they do not understand graph structure well (because of the prompts, or intrinsic limitations), or because this information is just not useful for the task for LLMs (because their good text understanding is enough). Some further investigations might be missing, e.g. try new prompts or focusing on text-scarce or even no text graphs!

That being said, even a “negative” conclusion can be very valuable, and I would understand that the paper focus on text-rich graphs if this is an important task in general. Coud the authors elaborate on this and / or revise some claims / title?

---

> ### Author Response · Authors · 2024-04-15
> **Response to Reviewer 5tBU**
>
> > “It seems to me that this work is in a sense incomplete: the title is "Can LLMs Effectively Leverage Graph Structural Information through Prompts, and Why?", but the conclusion is that LLMs do not seem to leverage this information for the particular task of study. Hence, it is not clear to me whether this is because they do not understand graph structure well (because of the prompts, or intrinsic limitations), or because this information is just not useful for the task for LLMs (because their good text understanding is enough).”
>
> Indeed, our study argues that for classification on text-attributed graphs, the structural information is less important than the text information. In Sec 3.3 we try various prompting templates and show that LLMs do not tend to understand the prompts in graph structure. And in Sec 3.4, we show that only the inclusion of text related to the target node can benefit the prediction. All in all, our study argues that the most efficient elements of the local neighborhood included in the prompt are phrases (node attribute) that are pertinent to the node label, rather than the graph structure. This corresponds to the “why” part of the question and gives a negative conclusion on how LLMs can leverage structural information on text-attributed graphs.
>
>
> > “Some further investigations might be missing, e.g. try new prompts or focusing on text-scarce or even no text graphs!”
>
> Thank you for your comment! We acknowledge that the focus of this paper is on text-attributed graphs, i.e. graphs that come with raw text attributes. We also note here text-attributed graphs can have different levels of text richness. We investigate the influence of text richness on prediction performance of LLMs in Sec. 3.5.
>
> The motivation for this paper is to provide a deeper understanding of recent studies on using LLMs for classification tasks on text-attributed graphs. We have added the “text-attributed graphs” into paper title to clarify the scope of this paper: “Can LLMs Effectively Leverage Graph Structural Information through Prompts **in Text-attributed Graphs**, and Why?”

---

> > ### Author Response · Authors · 2024-04-18
> > **Response to Reviewer 5tBU**
> >
> > Dear Reviewer 5tBU, thank you again for your effort in reviewing our paper. As the end of the discussion period approaches, we would like to ask if our responses were able to sufficiently address your concerns. If you have further questions, please let us know and we are eager to further address them!

---

### Author Response · Authors · 2024-04-15
**Comment for all**

We thank all reviewers for their insightful and constructive comments on our work. We are glad to see that our paper is commented to have answered “important questions” to probe LLMs' understanding of graph structures by **Reviewer UYwb and 5tBU**. Also, investigation of the potential impact of data leakage has been praised by **Reviewer UYwb and H17Y**. Lastly, the experimental results are recognized as “thorough” and “interesting” by **Reviewer H17Y**.

We have responded to individual questions below. We have also made a few modifications in our paper to make the purpose of this paper more clear, highlighted with the red font.

---

### Decision · Action_Editor_rZNs · 2024-05-31

**Recommendation:** Accept as is

**Comment:**

The paper was reviewed by three experts and received mixed ratings, with one reviewer recommending acceptance, one recommending weak acceptance and one rejection. The reviewers initially raised several concerns about misleading claims. For example, since the paper focuses only on text-attributed graphs, its findings cannot be generalized to other types of graphs. Moreover, no clear explanations were given about the relationship between homophily ratio and the performance of LLMs. Most of those concerns were addressed by the authors in the revision. I believe that the submission now meets the TMLR acceptance criteria, and thus, I am recommending acceptance.

**Audience:**

The findings of this paper will be of interest to some individuals in TMLR's audience.

**Claims And Evidence:**

The main claims made in the submission are the following: (1) LLMs' excellent performance in text-attributed graph representation learning tasks is not due to data leakage; (2) LLMs understand the input prompts as linearized paragraphs ; and (3) LLMs' predictive performance in text-attributed graph representation learning tasks generally increases as the local homophily ratio increases.

The above claims are supported by empirical evidence, while the employed experimental protocol seems convincing.